# A New Cretaceous Dustywing Genus (Neuroptera: Coniopterygidae) with Peculiar Wing Venation

**DOI:** 10.3390/insects13070654

**Published:** 2022-07-19

**Authors:** Zuluan Chen, Lihua Wang, De Zhuo, Chunpeng Xu, Xingyue Liu

**Affiliations:** 1Department of Entomology, China Agricultural University, Beijing 100193, China; chenzuluan@foxmail.com; 2Hubei Key Laboratory of Quality Control of Characteristic Fruits and Vegetables, College of Life Science and Technology, Hubei Engineering University, Xiaogan 432000, China; lwang@hbeu.edu.cn; 3Beijing Xiachong Amber Museum, 9 Shuanghe Middle Road, Beijing 100023, China; zhuode113@163.com; 4State Key Laboratory of Palaeobiology and Stratigraphy, Nanjing Institute of Geology and Palaeontology, Chinese Academy of Sciences, Nanjing 210018, China; cpxu@nigpas.ac.cn

**Keywords:** Coniopterygidae, Kachin amber, new genus, new species, morphology

## Abstract

**Simple Summary:**

The Cretaceous dustywings (Neuroptera: Coniopterygidae) are valuable for understanding the early evolution of this specialized lacewing family. Here, a new genus with two new species of Coniopterygidae is described from the mid-Cretaceous Kachin amber of northern Myanmar. The wing venation of the new genus is remarkably regarding the unique configuration of ScP, RA, and RP, the presence of forewing A3, and the distal gradate crossveins. *The new genus stands as a Cretaceous chimera among diverse dustywings by having a combination of plesiomorphic and apomorphic characters.*

**Abstract:**

The species and morphological diversity of dustywings (Neuroptera: Coniopterygidae) from the Cretaceous, of which the knowledge is rapidly increasing by recent studies on the species from the mid-Cretaceous Kachin amber, provide valuable evidence for understanding the phylogeny and early evolution of this highly specialized lacewing lineage. Here we describe a new genus and two new species of this genus in Coniopterygidae from the mid-Cretaceous (lowermost Cenomanian) of northern Myanmar, namely *Paradoxoconis szirakii* **gen. et sp. nov.** and *Paradoxoconis longipalpa* **gen. et sp. nov.** The new genus possesses a peculiar combination of wing characters, e.g., the terminal fusion or connection between ScP and RA, the terminal connection of RA to RP, the presence of forewing A3, and the presence of a distal gradate series of crossveins. Despite uncertain subfamilial placement, this new genus morphologically resembles the extant genus *Coniocompsa* Enderlein, 1905 of the subfamily Aleuropteryginae and the extant genus *Flintoconis* Sziráki, 2007 of the subfamily Brucheiserinae. Our finding highlights the palaeodiversity of dustywings from the Cretaceous.

## 1. Introduction

The lacewing family Coniopterygidae (dustywings), currently with ca. 580 extant species worldwide [1], represents a highly specialized group in Neuroptera by the minute body-size, the presence of waxy covering and the strong reduction of wing venation. The miniaturization and disputed phylogenetic position of Coniopterygidae have drawn much attention and invoked a number of studies on comparative morphology using new morphological techniques and on phylogeny using morphological and genomic data [2,3,4,5,6,7,8]. However, the evolutionary history of Coniopterygidae is still largely unknown. So far, there are three subfamilies (Aleuropteryginae, Brucheiserinae, and Coniopteryginae) in extant Coniopterygidae and an extinct subfamily (Cretaconiopteryginae) from the Cretaceous. Their subfamilial relationships and time-scale of divergences have not been recovered. The fossil records of Coniopterygidae serve as a pivotal source of data to trace the deep divergence and early evolution of dustywings. Hitherto, there have been 42 fossil dustywings described worldwide [9,10,11]. Among them, the mid-Cretaceous Kachin amber dustywings, with nine genera and 11 species described [9,12,13,14,15,16].

Here we report a new genus with two new species of Coniopterygidae from the mid-Cretaceous of northern Myanmar. The new genus, namely *Paradoxoconis*
**gen. nov.**, shows a set of peculiar wing characters and resembles the extant genera *Coniocompsa* Enderlein, 1905 (Aleuropteryginae) and *Flintoconis* Sziráki, 2007 (Brucheiserinae). The subfamilial placement of this new genus is discussed although it remains unclear.

## 2. Materials and Methods

The amber specimens for this study come from the Hukwang Valley in Tanai Township, Myitkyina District of Kachin State, (Yu et al. [17]: Figure 1), and are deposited in the Nanjing Institute of Geology and Palaeontology (NIGP), Chinese Academy of Sciences, Nanjing, and the Beijing Xiachong Amber Museum (BXAM), Beijing, and were legally collected that meet requirements about amber ethics [18]. The age of the amber deposit has been investigated and dated to be 98.8 ± 0.6 million years by UePb dating of zircons from the volcanoclastic matrix of the amber [19]. Photographs were taken by using LEICA M165C and LEICA S8APO stereo microscope systems with Nikon D850 digital camera. Wing and genital drawings were prepared by using LEICA M165C stereo microscope system. The figures were prepared with Adobe Photoshop CC 2019. 

Terminology of wing venation generally follows Aspöck et al. [20] and Li et al. [9]. We did not use the terminology of wing venation of Meinander [21] as it lacks homology with other lacewing families, although this wing venation system were frequently used in many papers on Coniopterygidae (e.g., Azar et al. [22], Engel [13]). The presently used vein nomenclature is given below with comparison of that used by Meinander [21] in corresponding parentheses. Terminology of male genitalia follows Aspöck and Aspöck [23] and Li et al. [9].

Abbreviations used for wing veins are as following (Terms in bracket are used in Meinander [21]): A, anal vein; C, costa; Cu, cubitus; CuA (Cu1), cubitus anterior; CuP (Cu2), cubitus posterior; M, media; MA (R4 ± 5), media anterior; MP (M), media posterior; R, radius; RA (R1), radius anterior; RP (R2 ± 3), radius posterior; ScP (Sc), subcostal posterior; scp-r, crossvein between ScP and R stem; scp-ra, crossvein between ScP and RA; ra-rp, crossvein between RA and RP; rp-ma, crossvein between RP and MA; r-mp, crossvein between R stem and MP; ra-rp ± ma, crossvein between RA and RP ± MA; rp ± ma-mp, crossvein between RP ± MA and MP; ma-mp, crossvein between MA and MP; mp1-mp2, crossvein between MP1 and MP2; mp-cua, crossvein between MP and CuA; cua-cup, crossvein between CuA and CuP; cup-a1, crossvein between CuP and A1; a1-a2, crossvein between A1 and A2.

## 3. Systematic Palaeontology


**Order Neuroptera Linnaeus, 1758**



**Family Coniopterygidae Burmeister, 1839**



**Subfamily incertae sedis**



**Paradoxoconis gen. nov.**


(Figure 1, Figure 2, Figure 3, Figure 4, Figure 5, Figure 6 and Figure 7)

Type species: *Paradoxoconis longipalpa* **sp. nov.** (by present designation)

**Diagnosis.** Large-sized dustywings (forewing length 4.9–5.5 mm). Antenna slenderly filiform (Figure 1D and Figure 5B), with 23–30 segments. Forewing about 3.0 times as long as wide; ScP and RA terminally fused with each other due to distal part of ScP abruptly bending toward RA (or alternatively interpreted as connected by a crossvein with RA) or anteriorly curved RA (Figure 2, Figure 3 and Figure 6); subcostal space distinctly widened, at least twice as wide as radial space (Figure 3 and Figure 6); distal part of RA connected with RP by a crossvein; RP ± MA originated at about proximal ¼–1/3; RP more or less curved anteriad and terminated at wing margin (Figure 3 and Figure 6); R and MP touching or connected by a short crossvein; three crossveins present between RP + MA and MP; MP lacking inflation and stiff setae, bifurcated nearly at midpoint; a gradate series of three or four crossveins present at wing apex between branches of RP, MA, MP and CuA (Figure 3 and Figure 6); A2 distally bifurcated; A3 present (Figure 3 and Figure 6). Hind wing venation similar to that of forewing at least for distal part. Abdominal plicatures present.

**Etymology.** The new generic name is a combination of “*paradox-*” (Greek, meaning “strange”, in reference to the peculiar wing venation in the new genus) and “*konis*” (Greek, meaning “dust”, a common suffix of the generic name of dustywings). The gender of the name is feminine. The genus is registered under urn:lsid:zoobank.org:act:EA8889B1-9650-4EED-8FCF-6B6A7578768E.


**Paradoxoconis szirakii sp. nov.**


(Figure 1, Figure 2, Figure 3 and Figure 4)

**Diagnosis.** Besides the generic diagnostic characters, the new species is also characterized by the following characters: frons well-sclerotized between antennal insertions; antenna with ca. 23 segments; terminal portion of subcostal space with acute angle toward RA; no radial crossvein connecting stem of RP ± MA; four crossveins near wing apex between branches of RP, MA, MP, and CuA forming a gradate series.

**Description of holotype.** Female. Body (Figure 1A,B) length 5.5 mm; integument pale yellowish brown.

Head (Figure 1D) nearly as long as wide, with large and prominent compound eyes; frons well-sclerotized between antennal insertions; vertex feebly domed. Antenna (Figure 1D,E) filiform, with ca. 23 segments; scape nearly twice as long as wide; pedicel slightly thinner and shorter than scape; flagellum much thinner than pedicel; each flagellomere slenderly subcylindrical, most flagellomeres each nearly three times as long as wide, but distal eight flagellomeres much shorter, each nearly as long as wide. Mouthparts not discernible.

Prothorax subquadrate, much narrower than head as well as than mesothorax. Legs slender, with dense short setae; tibiae slightly widened; tibial spurs absent; tarsus (Figure 1C) 5-segmented; tarsomere 1 longest, nearly equal to the combined length of tarsomeres 2 and 3; tarsomeres 3 and 4 distally expanded and flattened, but tarsomere 4 with much broader expansion; tarsomere 5 slightly shorter than tarsomere 1; pretarsal claws simple, proximally slightly produced; arolium absent.

Forewing (Figure 2A,B and Figure 3A,C) length 5.5 mm; membrane hyaline and immaculate; two subcostal veinlets present near wing base, basalmost one nearly vertical to ScP, but second one obviously inclined; two subcostal veinlets present near termination of ScP, disamost one inclined; ScP gradually approximating costal margin, terminally abruptly bending posteriad and touching RA (alternatively interpreted as ScP terminated at costal margin and distally connected with RA by an arcuate crossvein); basal subcostal crossvein not discernible due to poor preservation, distal subcostal crossvein present at subdistal portion of subcostal space; RA terminally connected by a short crossvein with RP, which is anteriorly curved and solely terminated at wing margin; origin of RP ± MA nearly at proximal 1/3; RP simple, distally curved anteriad; MA simple and straight; one rp-ma crossvein present; MP nearly touching R by a short oblique veinlet, distally bifurcated about at midpoint; three rp ± ma-mp crossveins present; one mp1-mp2 crossvein present; RP/MA fork nearly half length of MP1/MP2 fork; Cu separated near wing base; CuA and CuP both straight and simple; two mp-cua crossveins present; one cua-cup crossvein present; A1 simple; three cup-a1 crossveins present, two of them closely spaced and at basal section of CuP; A2 distally bifurcated into a long and a short simple branches; one a1–a2 crossvein present; A3 simple, with a crossvein running to wing margin; four crossveins near wing apex between branches of RP, MA, MP, and CuA forming a gradate series.

Hind wing (Figure 2C,D and Figure 3B,D) length 4.5 mm, immaculate; veins on proximal half not discernible; at least one subcostal veinlet present near termination of ScP, and another one subcostal veinlet present nearly at midpoint of costal space; ScP terminally abruptly bending posteriad and touching RA (alternatively interpreted as ScP terminated at costal margin and distally connected with RA by an arcuate crossvein); distal subcostal crossvein present at subdistal portion of subcostal space; RA terminally connected by a short crossvein with RP, which is anteriorly curved and solely terminated at wing margin; MA simple and straight; MP1 and MP2 with preserved distal sections simple and straight; CuA and CuP with preserved distal sections simple and straight; two simple anal veins preserved; four crossveins near wing apex between branches of RP, MA, MP, and CuA forming a gradate series.

Abdomen with plicature not discernible. Segment 8 (Figure 4) subtrapezoidal in lateral view, with ventral portion slightly produced posteriad. Gonocoxites 9 fused as a single tongue-shaped plate. Ectoproct unpaired, ovoid. An additional unpaired tongue-shaped sclerite present between ectoproct and fused gonocoxites 9, with apex exceeding gonocoxites 9.

**Type material.** Holotype. NIGP 180659 (amber piece preserving a nearly complete adult female of *P. szirakii*
**sp. nov.** and syninclusion of one cockroach, one beetle, and four dipterans; it is polished in the form of a nearly elliptical transparent cabochon, with length×width about 27.0 × 16.3 mm, height about 8.6 mm), Lowermost Cenomanian, Tanai Village, Hukawng Valley, northern Myanmar.

**Etymology.** The new species is dedicated to Dr. György Sziráki, who made tremendous contributions to the systematics of Coniopterygidae.


***Paradoxoconis longipalpa* sp. nov.**


(Figure 5, Figure 6 and Figure 7)

**Diagnosis.** Besides the generic diagnostic characters, the new species is also characterized by the following characters: frons depressed between antennal insertions; antenna with ca. 30 segments; terminal portion of subcostal space with acute angle toward ScP; one radial crossvein present at stem of RP ± MA; three crossveins near wing apex between branches of RP, MA, and MP forming a gradate series; distal-most mp-cua crossvein distinctly apart from and proximad former gradate crossveins.

**Description of holotype.** Female. Body (Figure 5A) length 5.7 mm; integument brown.

Head (Figure 5C) nearly as long as wide, with large and prominent compound eyes; frons slightly depressed between antennal insertions; vertex barely domed. Antenna filiform, with 30 segments; scape nearly twice as long as wide; pedicel nearly as large as scape; flagellum much thinner than pedicel; each flagellomere slenderly subcylindrical, nearly twice as long as wide. Mouthparts chewing-mandibulate, visible in lateral view; labrum slightly shorter than clypeus; mandibles largely concealed under labrum, short; maxilla with unsegmented galea narrowly blade-like, slightly longer than stipes; maxillary palpus (Figure 5C) strongly elongated, nearly five times as long as galea, and nearly twice as long as head length (distance between vertex to labrum), with five segments (palpomere 5 long rod-like, nearly as long as total length of palpomeres 1–4, palpomere 1 and 3 subequal in length, palpomere 2 and 4 subequal in length, both slight shorter than palpomere 1; palpomeres 1–4 each distally with a few short setae); labial palpus three-segmented, palpomere 3 largest and ovoid.

Prothorax slightly prolonged, much narrower than head as well as than mesothorax. Meso- and metathorax robust, each in lateral view nearly twice as long as wide. Legs slender, with many short setae; tibiae slightly widened; tibial spurs absent; tarsus (Figure 5D) 5-segmented; tarsomere 1 longest, nearly equal to combined length of tarsomeres 2 and 3; tarsomeres 3 and 4 distally expanded and flattened, but tarsomere 4 with much broader expansion; tarsomere 5 slightly shorter than tarsomere 1; pretarsal claws simple, proximally slightly produced; arolium absent.

Forewing (Figure 6A,B) length 4.9 mm; membrane hyaline and immaculate; two subcostal veinlets present near wing base, both vertical to ScP; no subcostal veinlets visible near termination of ScP; ScP gradually approximating and terminally ending into costal margin; one basal subcostal crossvein present, distal subcostal crossvein present at subdistal portion of subcostal space; RA distally curved anteriad and fused with ScP, and connected with RP by a short crossvein; another radial crossvein present at stem of RP ± MA; origin of RP + MA nearly at proximal 1/4; RP and MA, respectively, simple, nearly straight and feebly curved anteriad; one rp-ma crossvein present; MP nearly touching R by a short crossvein, distally bifurcated at about midpoint; three rp ± ma-mp crossveins present; one mp1-mp2 crossvein present; RP/MA fork nearly half length of MP1/MP2 fork; Cu separated near wing base; CuA and CuP both straight and simple; three mp-cua crossveins present; two cua-cup crossveins present; A1 simple; three cup-a1 crossveins present, spaced by subequal distance; A2 distally bifurcated into a long and a short simple branches; one a1-a2 crossvein present; A3 simple; three crossveins near wing apex between branches of RP, MA, MP forming a gradate series, but distal-most mp-cua distinctly proximad distal gradate crossveins.

Hind wing (Figure 6C,D) length 3.9 mm, immaculate; many veins on proximal half incompletely preserved or not discernible; no subcostal veinlets visible near termination of ScP; ScP gradually approximating and terminally ending into costal margin; distal subcostal crossvein present at subdistal portion of subcostal space; RA distally curved anteriad and fused with ScP, and connected with RP by a short crossvein; another radial crossvein present at stem of RP ± MA; origin of RP ± MA nearly at proximal 1/3; RP and MA, respectively, simple, nearly straight and feebly curved anteriad; one rp-ma crossvein present; MP distally bifurcated at about midpoint; two rp ± ma-mp crossveins present; one mp1-mp2 crossvein present; RP/MA fork slightly shorter than MP1/MP2 fork; Cu separated near wing base; CuA and CuP respectively simple; one mp-cua crossvein discernible; two cua-cup crossveins present; A1 with preserved part simple; one cup-a1 crossvein discernible; A2 distally bifurcated; one a1-a2 crossvein present; A3 probably present; three crossveins near wing apex between branches of RP, MA, MP forming a gradate series, but distal-most mp-cua distinctly proximad distal gradate crossveins.

Abdomen strongly attenuated from segment 6, laterally with four pairs of plicatures on segments 3–6 (Figure 5E,F). Segment 9 (Figure 7) seemingly fused as a ring-like structure, terminally with a strongly elongated sclerite, which in lateral view is arm-like, fused with serrate ectoproct on proximal-dorsal portion, and distally curved dorsad. 

**Type material.** Holotype. BXAM BA-NEU-010 (amber piece preserving a complete adult female of *P. longipalpa*
**sp. nov.** and syninclusion of a thrip; it is polished in the form of a nearly elliptical transparent cabochon, with length×width about 23.8 × 21.7 mm, height about 4.8 mm), Lowermost Cenomanian, Tanai Village, Hukawng Valley, northern Myanmar.

**Etymology.** The specific epithet “*longipalpa*” refers to the strongly elongated maxillary palpi in the new species. The species is registered under urn:lsid:zoobank.org:act:C673F0EB-92BC-4F60-B2EC-755E8B3AF9E3.

## 4. Discussion

*Paradoxoconis***gen. nov.** differs from all genera of Aleuropteryginae and Coniopteryginae by the presence of rp-ma and mp1-mp2 crossveins, and the presence of forewing A3. Moreover, the new genus undoubtedly does not belong to Coniopteryginae by lacking the subfamilial apomorphies, i.e., the distal origin of hind wing RP ± MA and the presence of single rp ± ma-mp crossvein. So far, the autapomorphies of Aleuroteryginae are unclear [15], but two apomorphic wing character states (the presence of stiff setae on stem of forewing MP and the closely spaced hind wing MP and CuA) shared by most aleuropterygines are absent in *Paradoxoconis*
**gen. nov.** However, it is notable that the new genus shares some particular wing characters with the aleuropterygine genus *Coniocompsa*, i.e., the terminal fusion between ScP and RA, and the distinctly widened forewing subcostal space (see Meinander [21]: Figure 45). The former character is described as Sc2/R1 bending forwards and striking Sc1 slightly before its end in Meinander [21] in terms of the interpretation of the distal subcostal crossvein as the second branch of Sc. No matter how to interpret the venational homology, the configuration of the distal parts of ScP and RA is remarkable in the family. *Paradoxoconis longipalpa*
**sp. nov.** shows more similarity with Coniocompsa due to the anteriorly curved RA. In *P. szirakii*
**sp. nov.** the fusion between ScP and RA seems to be formed by the ScP terminally abruptly bending posteriad and touching RA. Moreover, the new genus and Coniocompsa both have similar female genital characters, particularly the large concaved sclerite protruded from the segment 9 (see Meinander [21]: Figures 45 and 46). Meinander [21] interpreted this structure as sternite 10 (the dorsal part) and gonapophyses laterales (the ventral part). Following the terminology proposed in Aspöck and Aspöck [23], the ventral part should represent the gonocoxites 9, while the homology of the dorsal part remains unknown. The aforementioned characters shared by *Paradoxoconis*
**gen. nov.** and *Coniocompsa* appear to be apomorphic and suggest that the new genus may be a stem-group Coniocompsini in Aleuropteryginae.

*Paradoxoconis***gen. nov.** shares some wing characters with Cretaconiopteryginae, such as the terminal connection of RA to RP, and the presence of a distal gradate series of crossveins. However, the new genus can be distinguished from Cretaconiopteryginae by the RA and RP not forming a loop, the smooth forewing CuP, the presence of a long forewing A3, and the greatly differed female genital characters, which may not support the placement of the new genus in Cretaconiopteryginae. 

Notably, the new genus resembles the extant genus *Flintoconis* of Brucheiserinae, which is considered to be the sister group of Coniopteryginae based on two larval apomorphic characters by Zimmermann et al. [2], in having similar configuration of forewing anal veins. *Flintoconis*, currently with only two species, both endemic to Chile, greatly differs from the other brucheiserine genus *Brucheiser* Navás, 1927 by the long membranous wings (wings long and sclerotized in the latter genus), although these two genera share the relatively small head dorsally covered by the large pronotum, the presence of plicatures, and some genital characters [24]. *Flintoconis* is the only extant dustywing genus known to date having the third anal vein (A3) in the forewing (considered as jugal vein by Sziráki [24]), while *Paradoxoconis*
**gen. nov.** is hitherto the only fossil dustywing with the forewing A3. If considering the general venational reduction in Coniopterygidae as apomorphic condition, the presence of forewing A3 may be interpreted as a plesiomorphic character state that remains in certain basal dustywings, and may not support any close phylogenetic relationship between *Paradoxoconis*
**gen. nov.** and *Flintoconis*. Moreover, many other wing characters greatly differ between these two genera, such as the configuration of ScP, RA, and RP, and the presence/absence of distal gradate series of crossveins. The head is much larger and not covered dorsally by the pronotum in *Paradoxoconis*
**gen. nov.**, while the head is smaller and retracted under the pronotum in *Flintoconis* (a subfamilial diagnosis of Brucheiserinae).

As discussed above, there has been no strong evidence to determine the subfamilial affiliation of *Paradoxoconis*
**gen. nov.** Nevertheless, the new genus stands a Cretaceous chimera among diverse dustywings by having a combination of plesiomorphic and apomorphic characters. The plesiomorphic characters comprise the basal origin of RP ± MA in both fore- and hind wings, the presence of forewing A3, and the dense crossvenation, while the apomorphies include the strongly elongated maxillary palpi, the highly modified ScP and RA, the strongly widened subcostal space, and the large concaved sclerite of the female genitalia. Future phylogenetic analysis combining fossil and extant dustywings may recover the placement of this enigmatic genus in the family.

## 5. Conclusions

In conclusion, *Paradoxoconis*
**gen. nov.** from the mid-Cretaceous of Myanmar represents a hitherto unknown lineage of dustywings with unclear subfamilial placement by having peculiar wing characters. The finding highlights the species and morphological diversity of the Cretaceous dustywings.

## Figures and Tables

**Figure 1 insects-13-00654-f001:**
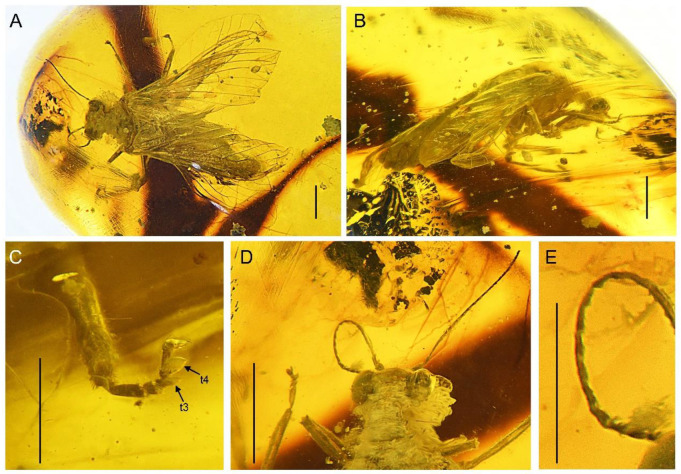
*Paradoxoconis szirakii* **gen. et sp. nov.**, holotype: NIGP 180659. (**A**) Habitus, dorsal view; (**B**) habitus, lateral view; (**C**) metatarsus; (**D**) head and pronotum, dorsal view; (**E**) left antenna. Abbreviation: t, tarsomere. Scale bar = 1.0 mm (**A**,**B**,**D**); 0.5 mm (**C**).

**Figure 2 insects-13-00654-f002:**
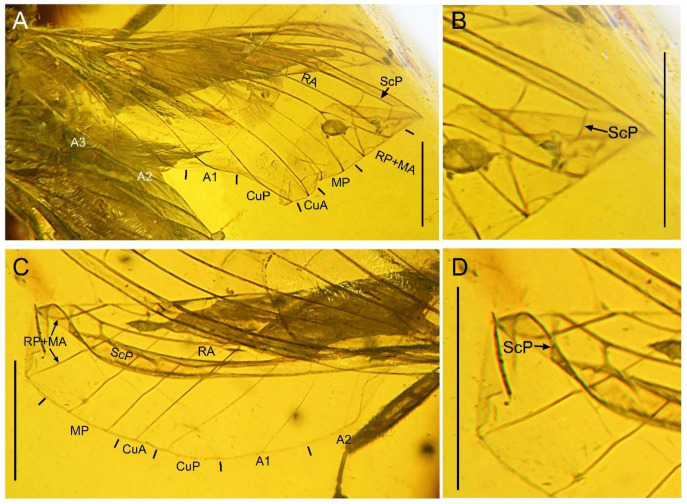
*Paradoxoconis szirakii* **gen. et sp. nov.**, holotype: NIGP 180659. (**A**) Right forewing; (**B**) antero-distal part of right forewing; (**C**) right hind wing; (**D**) antero-distal part of right hind wing. Scale bar = 1.0 mm.

**Figure 3 insects-13-00654-f003:**
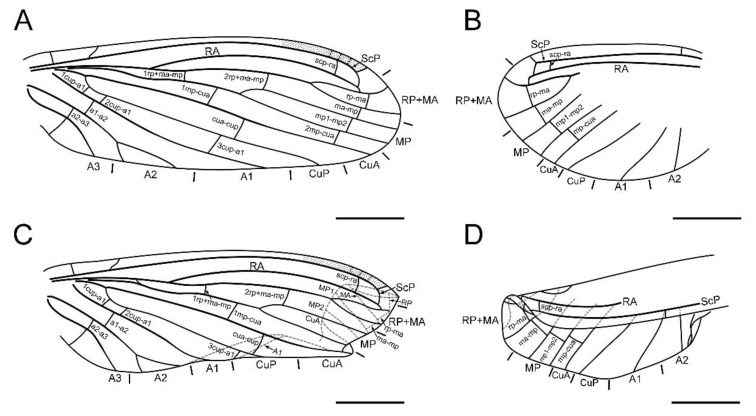
*Paradoxoconis szirakii* **gen. et sp. nov.**, holotype: NIGP 180659, drawings of wing venation. (**A**) Right forewing, with partial reconstruction; (**B**) right hind wing, with partial reconstruction; (**C**) right hind wing; (**D**) right hind wing, with partial reconstruction. Scale bar = 1.0 mm.

**Figure 4 insects-13-00654-f004:**
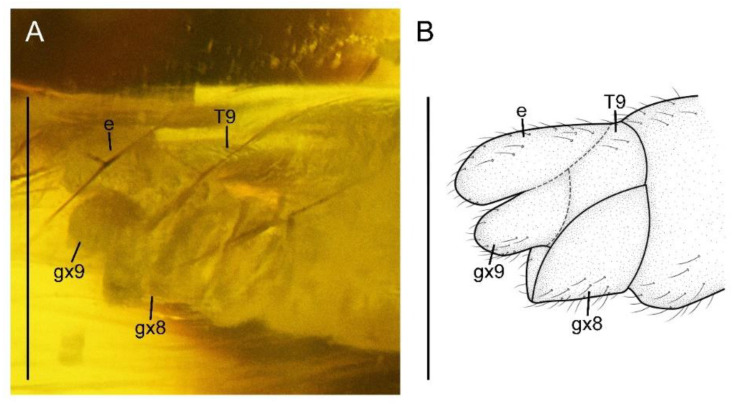
*Paradoxoconis szirakii* **gen. et sp. nov.**, holotype: NIGP 180659. (**A**) Female genitalia, lateral view; (**B**) female genitalia, line drawing, lateral view. Abbreviation: e, ectoproct; gx, gonocoxite; T, tergum. Scale bar = 1.0 mm.

**Figure 5 insects-13-00654-f005:**
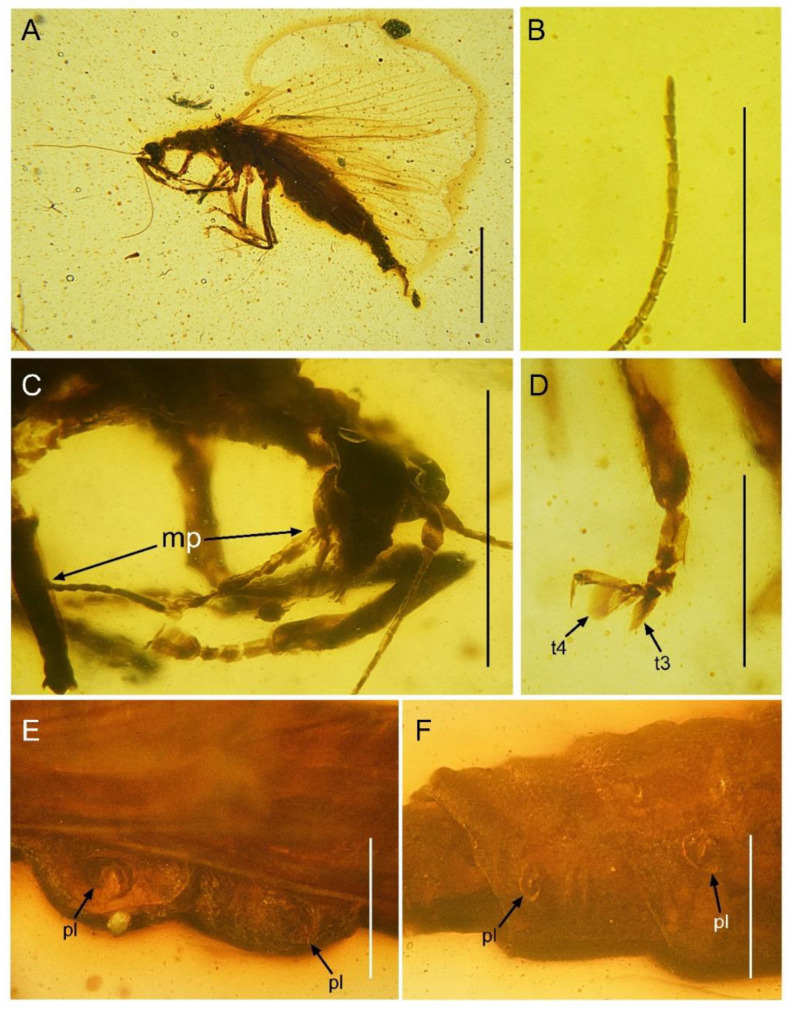
*Paradoxoconis longipalpa* **gen. et sp. nov.**, holotype: BXAM BA-NEU-010. (**A**) Habitus, lateral view; (**B**) apical flagellomeres; (**C**) head, lateral view; (**D**) metatarsus; (**E**) plicatures on left side of abdomen; (**F**) plicatures on right side of abdomen. Abbreviation: mp, maxillary palpus; t, tarsomere; pl, plicature. Scale bar = 1.0 mm (**A**); 0.5 mm (**C**); 0.25 mm (**B**,**D**,**E**,**F**).

**Figure 6 insects-13-00654-f006:**
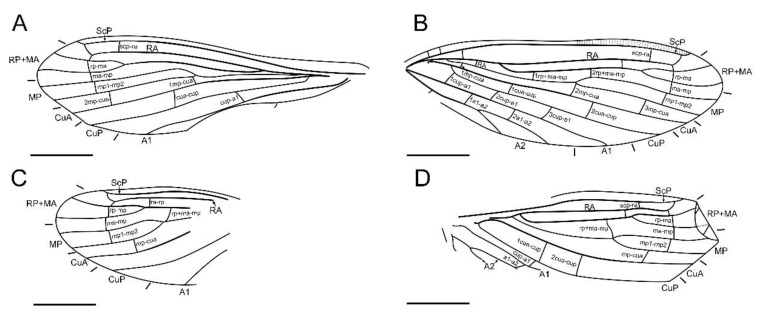
*Paradoxoconis longipalpa* **gen. et sp. nov.**, holotype: BXAM BA-NEU-010, drawings of wing venation. (**A**) Right forewing; (**B**) left forewing; (**C**) right hind wing; (**D**) left hind wing. Scale bar = 1.0 mm.

**Figure 7 insects-13-00654-f007:**
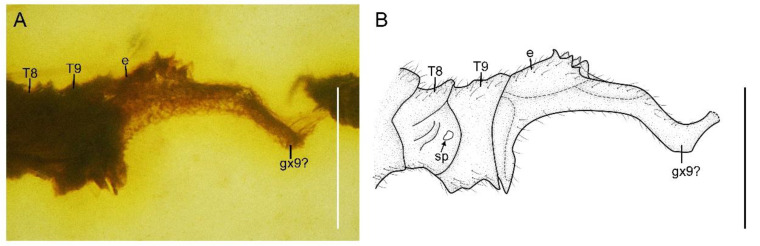
*Paradoxoconis longipalpa* **gen. et sp. nov.**, holotype: BXAM BA-NEU-010. (**A**) Female genitalia, lateral view; (**B**) female genitalia, line drawing. Abbreviation: e, ectoproct; gx, gonocoxite; sp, spiracle; T, tergum. Scale bar = 0.25 mm.

## Data Availability

All data are provided in the manuscript.

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
