# Peer review of "A New Cretaceous Dustywing Genus (Neuroptera: Coniopterygidae) with Peculiar Wing Venation"

_insects, 2022, doi:10.3390/insects13070654_

Round 1
Reviewer 1 Report
Chen et al. described a new genus with two new species of Coniopterygidae from Myanmar amber and highlights the palaeodiversity of dustywings from the Cretaceous. The taxonomy affinity of the new species and genus was well demonstrated. The only problem is that the authors should emphasize that their materials are legally obtained that meet recent requirements about Amber ethics; I believe this is no problem for this study but it's better to cite relevant reference that support legally study of materials from Myanmar (ref: Shi, C., Cai, H. H., Jiang, R. X., Wang, S., Engel, M. S., Yuan, J., ... & Spicer, R. A. (2021). Balance scientific and ethical concerns to achieve a nuanced perspective on ‘blood amber’. Nature Ecology & Evolution, 5(6), 705-706.)
Reviewer 2 Report
In general a nice and well written paper. Only few things might be improved:
Line 52: remove excess of italics
Line 88: this is a especialized journal called INSECTS you don't need to overcharge the paper including the class.
Line 95: Italics on the scientific name, and add also "by present designation"
Finally a side comment, the resemblance with Flintoconis will be nice to study further in the future, as many Chilean Neuroptera are endemic and usually primitive - maybe descendant of Araucaria associated Neuroptera; which might be the case of some Burmite Neuropterans.
